# Translation-equivariant Representation in Recurrent Networks with a Continuous Manifold of Attractors

**Wen-Hao Zhang**[1,2*]
wenhao.zhang@utsouthwestern.edu

**Ying Nian Wu**[3]
ywu@stat.ucla.edu

**Si Wu**[4]
siwu@pku.edu.cn

[1]Lyda Hill Department of Bioinformatics, UT Southwestern Medical Center.
[2]O'Donnell Brain Institute, UT Southwestern Medical Center.
[3]Department of Statistics, UCLA.
[4]School of Psychology and Cognitive Sciences, IDG/McGovern Institute for Brain Research, Center of Quantitative Biology, Peking-Tsinghua Center for Life Sciences, Peking University.

## Abstract

Equivariant representation is necessary for the brain and artificial perceptual systems to faithfully represent the stimulus under some (Lie) group transformations. However, it remains unknown how recurrent neural circuits in the brain represent the stimulus equivariantly, nor the neural representation of abstract group operators. The present study uses a one-dimensional (1D) translation group as an example to explore the general recurrent neural circuit mechanism of the equivariant stimulus representation. We found that a continuous attractor network (CAN), a canonical neural circuit model, self-consistently generates a continuous family of stationary population responses (attractors) that represents the stimulus equivariantly. Inspired by the Drosophila's compass circuit, we found that the 1D translation operators can be represented by extra speed neurons besides the CAN, where speed neurons' responses represent the moving speed (1D translation group parameter), and their feedback connections to the CAN represent the translation generator (Lie algebra). We demonstrated that the network responses are consistent with experimental data. Our model for the first time demonstrates how recurrent neural circuitry in the brain achieves equivariant stimulus representation.

## 1 Introduction

Neuronal responses in the brain internally represent the stimulus $s$ in the world. When the stimulus changes due to some transformations, the brain updates its internal representation accordingly to faithfully reflect the stimulus change. For example, V1 neurons encode the visual orientation, and their population responses track the changing orientation which enables subsequent brain areas to decode the instantaneous orientation. Transformations acting on a continuous stimulus can be described by the Lie group theory [1], and a neural representation reflecting the stimulus transformation is called group *equivariant* representation [2–4]. Mathematically, supposing a Lie group $G$ acting on a continuous stimulus $s$, the evoked neural population response $\bar{u}(s)$ (a continuous function of $s$, and $\bar{u}$ denotes the mean response) is equivariant with the action of group $G$ (group *homomorphism*) if,

$$\hat{T}_g \cdot \bar{u}(s) = \bar{u}(T_g \cdot s), \quad \forall g \in G, \forall s. \tag{1}$$

---

*Corresponding author.

36th Conference on Neural Information Processing Systems (NeurIPS 2022).

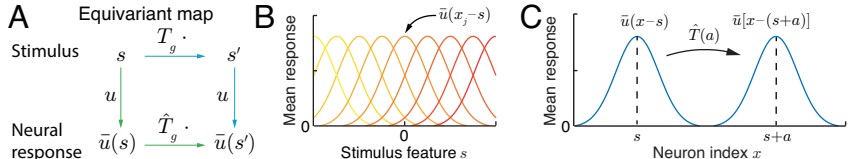

Figure 1: (A) An equivariant map between the stimulus and the neural representation. (B) Tuning curves (mean responses) of a population of homogeneous neurons in response to a one-dimensional stimulus feature $s$, where each neuron $j$ has its unique value of preferred stimulus $x_j$. The tunings of neurons have the same profile and only differ in the preferred stimulus $x_j$. (C) A one-dimensional translation operator $\hat{T}(a)$ shifts the population response $\bar{u}(x - s)$ by amount $a$.

$\hat{T}_g$ and $T_g$ are the induced actions (operators) with group element $g \in G$ which respectively act on neuronal responses and the stimulus, and $\cdot$ denotes the action of group operators. Intuitively, the equivariant neural representation (Eq. 1) implies the neuronal responses evoked from a transformed stimulus (e.g., rotating the orientation of Gabor images), i.e., $\bar{u}(T_g \cdot s)$ (Fig. 1A, blue path) would be the same as transforming the neuronal responses evoked by the original stimulus along the stimulus manifold, i.e., $\hat{T}_g \cdot \bar{u}(s)$ (Fig. 1A, green path).

The equivariant representation is indispensable for the brain to represent the stimulus under transformations. An equivariant neural circuit is not only required to represent the stimulus equivariantly ($\bar{u}(s)$, Eq. 1), but it also needs to *explicitly* represents the Lie group operators ($\hat{T}_g$, Eq. 1). Explicitly representing group operators enables the brain to actively synthesize a transformed stimulus and predict the changing stimulus under transformations [5]. For example, when tracking a moving object, the brain not only represents the object's instantaneous position $s$, but also represents its moving speed (the translation group parameter) in order to predict the stimulus position in the next time step. Yet it is an open question as to how recurrent neural circuits in the brain equivariantly represent the stimulus, and how abstract group operators are represented in the brain.

Neuroscientists utilized the Lie group to investigate the perception and representation in the brain (e.g., [6–14]), however, nearly none of previous studies linked the abstract group theory to concrete biological recurrent circuit dynamics. Meanwhile, although equivariant neural networks have been extensively studied recently in machine learning (ML) research to capture transformations of stimuli (input data) defined on graphs (e.g., [15]) or defined on smooth manifolds (e.g., [3, 4, 16, 17]), nearly all these studies focused on developing feedforward weights which act as group convolution kernels in order to yield equivariant outputs under group transformations. There has not been much ML research studying equivariant recurrent networks, nor explicit network representations of group operators.

To provide theoretical insight into this fundamental question, we used a 1D translation group acting on a 1D stimulus feature as an example to investigate the general recurrent circuit mechanism of equivariant representation. We found the equivariant stimulus representation can be naturally realized in continuous attractor networks (CANs) which self-consistently generate a continuous family of stationary population responses (attractors) to represent the stimulus equivariantly. Moreover, the translation group operators are represented by separate neuronal populations whose responses represent the translation speed (translation group parameter). The feedback connections from speed neurons to CANs are shifted on the stimulus manifold, which encode the translation *generator* (Lie algebra). We compare the proposed recurrent circuit model with Drosophila's internal compass circuit to show its biological plausibility. The present study is one of the first studies linking the equivariant representation to a biologically plausible neural circuit model with rigorous mathematical analysis. It will shed light on the neural representations in the brain.

## 2    Translation group and homogeneous neural population code

We start from a concrete Lie group and neural representation to investigate the general circuit mechanism of equivariant representation. As an example, we consider a 1D translation Lie group $\mathbb{T}$ acting on a 1D stimulus $s$ which can be regarded to be heading direction, orientation, moving direction, spatial location, etc. A group element $T(a) \in \mathbb{T}$ translates the stimulus $s$ to $s + a$, i.e., $T(a) \cdot s = s + a$, and therefore an equivariant neural representation $\bar{u}(s)$ should satisfy (Fig. 1C),

$$\bar{u}\big[T(a) \cdot s\big] = \bar{u}(s + a) = \hat{T}(a) \cdot \bar{u}(s), \tag{2}$$

where $\hat{T}(a)$ is the 1D translation operator inducing the translation of neural representation (Fig. 1C). Several properties of the $\hat{T}(a)$ can be directly derived from the definition (Eq. 2) [18],

$$\hat{T}(0) = 1, \tag{3a}$$

$$\hat{T}(a)\hat{T}(b) = \hat{T}(a + b) = \hat{T}(b)\hat{T}(a), \tag{3b}$$

$$\hat{T}(a)^{-1} = \hat{T}(-a). \tag{3c}$$

First, a translation operator with zero translation should leave the neural representation of stimulus to be the same as no transformation (Eq. 3a). Second, the translation operator can be composed together and the composition is not dependent on the order of operators (Eq. 3b), which implies the translation operator is *commutative* or *abelian* [18]. Third, the inverse of a translation operator corresponds to an operator translating to the opposite direction with the same distance (Eq. 3c). Since the translation is continuous, the amount of translation can be made infinitesimally small. Consider an infinitesimal translation $\delta a \to 0$ on the stimulus $s$, then the corresponding change of neural representation is,

$$\hat{T}(\delta a) \cdot \bar{u}(s) = \bar{u}(s + \delta a) \approx \bar{u}(s) + \delta a \partial_s \bar{u}(s) = (1 + \delta a \partial_s)\bar{u}(s), \tag{4}$$

where $\partial_s = \partial/\partial s$ denotes the derivative over $s$, and a first order Taylor expansion was used in above derivation. $\hat{p} \equiv \partial_s$ is called the translation *generator* characterizing the tangential direction of translation in the group space, and forms a basis of the Lie *algebra*. In physics, $\hat{p}$ is usually regarded as the momentum operator [18]. Using the infinitesimal translation, the differential form of a translation operator can be derived as (see details of Lie group in Supplementary Information (SI)),

$$\frac{d\hat{T}(a)}{da} = \lim_{\delta a \to 0} \frac{\hat{T}(a + \delta a) - \hat{T}(a)}{\delta a} = \left(\lim_{\delta a \to 0} \frac{\hat{T}(\delta a) - 1}{\delta a}\right)\hat{T}(a) = \hat{p} \cdot \hat{T}(a). \tag{5}$$

By exponentiating the two sides in Eq. (5), the translation operator can be solved as $\hat{T}(a) = \exp(a\hat{p})$, which is an exponential map of the translation generator $\hat{p}$.

We next define a concrete neural representation based on experimental observations. The neural representation $\bar{u}(s) = \{\bar{u}_j(s)\}_{j=1}^N$ is regarded as the mean responses of a population of $N$ neurons. Since the translation operator (Eq. 2) acts on a continuous function of neural population response, i.e., $\bar{u}(s)$, we therefore consider the limit of a large neuronal population with infinite number of neurons ($N \to \infty$), and then $\bar{u}(s)$ effectively converges to a continuous function (a vector of infinite dimension in Hilbert space). Inspired by the bell-shape stimulus tuning curves widely observed in the cortex, the mean response of the $j$-th neuron is modeled as a Gaussian function of the stimulus $s$ (Fig. 1B),

$$\bar{u}_j(s) \equiv \bar{u}(x_j - s) = \mathsf{U}_j \exp[-(x_j - s)^2/4\sigma_j^2], \tag{6}$$

where $x_j$ denotes the preferred stimulus of a neuron, indicating the neuronal response reaches its maximum when the stimulus matches its preferred value, i.e., $s = x_j$. $\mathsf{U}_j$ and $\sigma_j$ denote respectively the peak firing rate and the tuning width of the $j$-th neuron. A necessary condition to construct an equivariant neural population code is *homogeneous* neurons in the population, which means the tuning curves of all neurons have the same profile, i.e., $\mathsf{U}_j \equiv \mathsf{U}$ and $\sigma_j \equiv \sigma$ for any $j$, and the preferred stimulus of all neurons, $\{x_j\}$, is uniformly distributed on the stimulus space. In this way, the mean responses of all neurons in response to a stimulus $s$ are (Fig. 1C)

$$\bar{u}(s) \equiv \bar{u}(x - s) = \mathsf{U} \exp[-(x - s)^2/4\sigma^2], \tag{7}$$

which can be verified to satisfy the translation equivariant representation (Eq. 2).

## 3 Towards an equivariant recurrent neural circuit model

We next investigate how the equivariant homogeneous neural code, i.e., $\bar{u}(x - s)$ in Eq. (7), can be self-consistently generated by a concrete recurrent neural circuit dynamics. Specifically, we explore the recurrent circuit mechanism to generate a stable stimulus representation $\bar{u}(x - s)$ without translation (corresponding to a zero translation operator, $\hat{T}(0)$). Afterward, we investigate the neural representation of translation operators and how it induces the translation of stimulus representation.

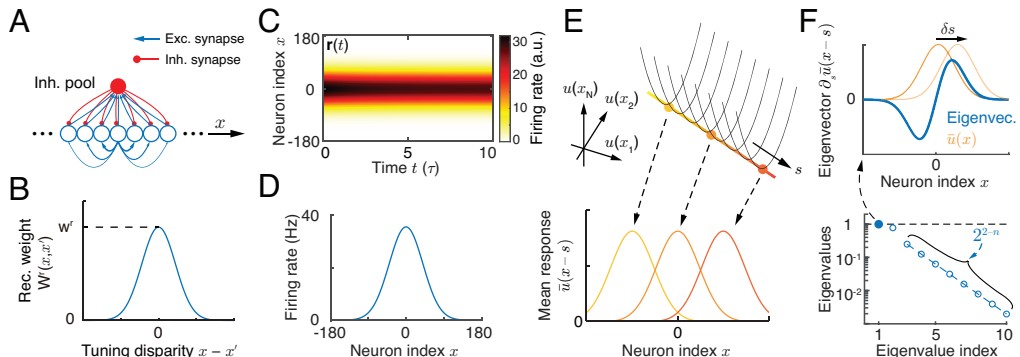

Figure 2: A continuous attractor network (CAN) self-consistently maintains an equivariant stimulus representation. (A) The CAN structure. Excitatory (E) neurons are aligned based on their preferred stimulus $x$, and a pool of inhibitory neurons provides negative feedback to ensure the network stability. (B) The recurrent connections between E neurons are translation-invariant with respect to the preferred stimulus $x$. (C-D) An example of spatiotemporal responses of all E neurons in the network model (C) and their temporal average (D). Note that the non-zero response is self-sustained by the network without external input. (E) The state space of CAN's population responses contains a one-dimensional continuous manifold, on which each location denotes a population response $\bar{u}(x-s)$ representing the value of stimulus feature $s$. Top: a schematic diagram of the energy landscape of the CAN. Bottom: three example population responses which correspond to three states on the continuous manifold. (F) Eigenanalysis of the perturbed CAN dynamics. Top: the eigenvector (blue) with the largest eigenvalue ($\lambda_1 = 1$) corresponds to an infinitesimal change of population responses (orange) along the stimulus feature manifold. Bottom: the eigenvalues of the perturbed CAN dynamics where the largest eigenvalue is 1, indicating neutral stability along the stimulus manifold.

## 3.1 The autonomous dynamics of continuous attractor networks

Since the recurrent neural circuit dynamics needs to generate non-zero mean responses $\bar{u}(x-s)$ for any value $s$ (Eq. 6), the neural dynamics should be stable at any $s$. That is, there is a manifold of fixed points (attractors) in the state space of network dynamics. The continuous manifold of attractors can emerge from the autonomous dynamics (no external input drive) of continuous attractor networks (CANs, Fig. 2A), which is a canonical network model explaining the processing of continuous stimuli in neural circuits [19, 20], including heading-direction [21], orientation [22], spatial location [23] and so on. In the present study, we consider a CAN dynamics as follows [24, 25],

$$\tau \frac{\partial u(x,t)}{\partial t} = -u(x,t) + \rho \int \mathsf{W_r}(x,x') \cdot r(x',t)dx', \tag{8a}$$

$$r(x,t) = G[u(x,t)] = \frac{[u(x,t)]_+^2}{1 + k\rho \int [u(x',t)]_+^2 dx'}. \tag{8b}$$

where $u(x,t)$ and $r(x,t)$ denote respectively the synaptic input and firing rate of the excitatory neuron preferring stimulus $s = x$ at time $t$, and $\tau$ is the time constant of synaptic input. $G(\cdot)$ is an activation function that is modeled as divisive normalization where $[\cdot]_+$ is nonnegative rectification and $k$ determines the global inhibition strength. The divisive normalization is a canonical operation widely observed in the cortex [26], and is resulted from the parvalbumin (PV) inhibitory neurons [27].

$\mathsf{W_r}(x,x')$ denotes the recurrent weight from the neuron preferring $x'$ to the neuron preferring $x$, which is modeled as a Gaussian function of the disparity between the preferred stimuli of the pre- and post-synaptic neurons (Fig. 2B), i.e.,

$$\mathsf{W_r}(x,x') = \mathsf{W_r}(x-x') = \mathsf{w_r}(\sqrt{2\pi}\sigma)^{-1} \exp\left[-(x-x')^2/2\sigma^2\right], \tag{9}$$

where $\mathsf{w_r}$ is a scalar and determines the peak recurrent weight, and $\sigma$ the width of connection in the stimulus space. Because we consider a large population of neurons that are uniformly distributed in the stimulus space, the neuronal interactions effectively become an integral in Eq. 8a, with $\rho$ denoting the neuronal density in the stimulus space. Furthermore, the translation-invariant recurrent connections, which only depend on $x - x'$, imply that the recurrent interactions are effectively

conducting *convolution* on the stimulus space, i.e., $\int \mathsf{W_r}(x-x') \cdot r(x',t)dx' \equiv (\mathsf{W_r} * r)(x,t)$, where the symbol $*$ denotes the convolution over $x$.

## 3.2 A continuous attractor network equivariantly represents the stimulus

It can be checked that the CAN dynamics (Eqs. 8a-9) can generate a family of non-zero Gaussian-profile stationary responses that are consistent with the proposed responses (Eq. 6) [25, 28],

$$\bar{u}(x-s) = \mathsf{U}\exp\left[-(x-s)^2/4\sigma^2\right], \quad \bar{r}(x-s) = \mathsf{R}\exp\left[-(x-s)^2/2\sigma^2\right], \qquad (10)$$

as long as the recurrent weight is stronger than a critical value which is solved as $\mathsf{w_c} = 2\sqrt{2}(2\pi)^{1/4}\sqrt{k\sigma/\rho}$ (see SI. Sec. 2). It is worth noting that $s$ is a free parameter in Eq. (10), indicating that the network has a continuous family of fixed points, i.e., $\bar{u}(x-s)$, which self-consistently emerge from the recurrent neural dynamics without the influence from external inputs (Eqs. 8a-8b). The stationary population responses (Eq. 10) are consistent with the equivariant homogeneous neural population code (Eq. 7), and therefore they represent the stimulus equivariantly.

The stationary population responses in the recurrent network should be stable in order to maintain the equivariant stimulus representation robustly, otherwise, the population responses can be easily disrupted by noises. Therefore we check whether the continuous family of population responses (Eq. 10) are stable (attractors), by performing the stability analysis of the network dynamics. We perturbed the network state around the fixed points by adding a small perturbation $\delta u(x,t)$, i.e., $u(x,t) = \bar{u}(x-s) + \delta u(x,t)$, and then the dynamics of the perturbation is derived as [28],

$$\tau\frac{\partial}{\partial t}\delta u(x,t) = -\delta u(x,t) + \int K(x,x'|s)\delta u(x',t)dx'. \qquad (11)$$

The interaction kernel $K(x,x'|s) = \rho\int \mathsf{W_r}(x-x'')\partial\bar{r}(x''-s)/\partial\bar{u}(x'-s)dx''$, whose analytical expression can be seen at SI. Sec. 3. Treating $K(x,x'|s)$ as an operator acting on the perturbation $\delta u(x,t)$, all of its eigenvalues, $\lambda_n$, and (unnormalized) eigenfunctions, $f_n$, can be analytically calculated (listed with a descending order) [28],

$$\lambda_1 = 1, \qquad\qquad f_1(x|s) \propto \partial_s\bar{u}(x-s), \qquad (12a)$$

$$\lambda_2 = 1 - \sqrt{1 - \mathsf{w_c}^2/\mathsf{w_r}^2}, \quad f_2(x|s) \propto \bar{u}(x-s), \qquad (12b)$$

$$\lambda_n = 2^{2-n} \ (n \geq 3), \qquad f_n(x|s) = \cdots. \qquad (12c)$$

The explicit expression of eigenfunctions for $n \geq 3$ can be found at SI. Sec. 3. We see no eigenvalues are larger than 1 (Fig. 2F) indicating that the continuous family of fixed points ($\bar{u}(x-s)$ in Eq. 10) are stable, i.e., the network has a continuous family of attractors, and each attractor is an equivariant representation of the stimulus $s$ (Fig. 2E).

# 4 Representing translation operators in CANs

It is noteworthy that the eigenfunction $f_1(x|s)$ of the perturbed network dynamics (Eq. 2F, top) is parallel to the direction of translation generator (Eq. 12a), i.e., $f_1(x|s) \propto \partial_s\bar{u}(x-s) = \hat{p} \cdot \bar{u}(x-s)$, thus $f_1(x|s)$ corresponds to the translation of neural responses along the continuous attractor manifold representing the stimulus $s$ (Eq. 2F, top). Meanwhile, its corresponding eigenvalue $\lambda_1 = 1$ suggests the eigenfunction $f_1(x|s)$ is *neutrally* stable, which is resulted from the translation invariance of recurrent connections in the network (Eq. 9). Nevertheless, although the CAN dynamics maintains the translation operator resulted from the external perturbation $\delta u(x,t)$, it does not self-consistently support non-zero translation generators in that the stationary network states are static without any translation (Eq. 10, Fig. 2C-D), i.e., corresponding to $\hat{T}(0)$. We next explore a self-sustained network mechanism to represent non-zero translation operators, i.e., $\hat{T}(a)$, $(a \neq 0)$ (Eq. 2), in order to induce translations of neuronal responses $\bar{u}(x-s)$.

## 4.1 Incorporating translation operators in a CAN

To embed translation operators in the CAN dynamics, we derive a differential equation of Eq. (2) and compare it with the CAN dynamics. We consider the translation has instantaneous speed $v(t)$, which determines the translation amount in a time window $dt$ as $da = v(t)dt$. Taking the derivative of Eq.

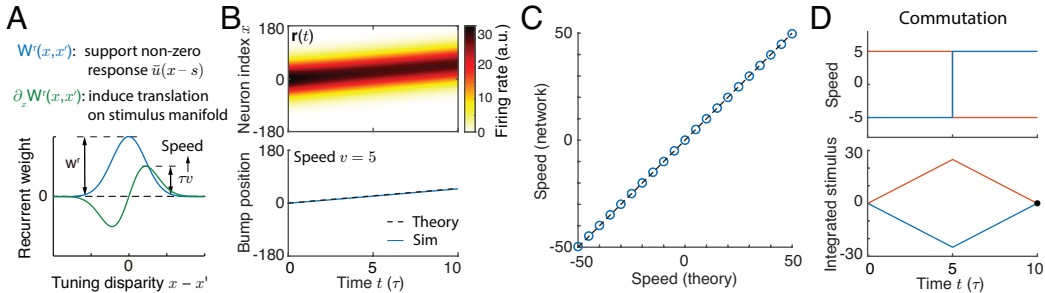

Figure 3: The derivative of recurrent connection, $\partial_x W_r$, induces translation of CAN's population responses along the stimulus feature manifold, and represents the 1D translation operator. (A) Illustration of the recurrent connections in the network model. (B) An example of spatiotemporal responses in the network (top) and the internally represented stimulus feature moves over time with a constant speed. (C) The actual moving speed of population responses along the stimulus manifold is consistent with theoretical prediction by using the magnitude of $\partial_x W_r$. (D) Commutative path integration in the CAN.

(2) over time $t$, and utilizing the differential form of translation operators (Eq. 5), we derive that (to simplify notation we set $s = 0$ without loss of generality),

$$\frac{d\bar{u}(x-a)}{dt} = \frac{d\hat{T}(a)\cdot\bar{u}(x)}{da}\frac{da}{dt} = v\left[\frac{d\hat{T}(a)}{da}\bar{u}(x)\right] = v\hat{p}\cdot\hat{T}(a)\cdot\bar{u}(x) = v\hat{p}\cdot\bar{u}(x-a).$$

Replacing the variable $x - a$ by $x$, we reach the differential equation governing the translation of neuronal responses with speed $v(t)$,

$$\frac{d\bar{u}(x)}{dt} = v(t)\hat{p}\cdot\bar{u}(x). \tag{13}$$

In comparison, the CAN dynamics in the stationary state is $d\bar{u}(x)/dt = 0$ (Eq. 8a), i.e., the network state is static. To incorporate the translation operator with speed $v(t)$ (Eq. 13), we utilize that the stationary CAN state satisfying $\bar{u}(x-s) = \rho W_r * \bar{r}(x-s)$, and substitute it to the right hand side of Eq. (13),

$$v\hat{p}\cdot\bar{u}(x) = v\partial_s[\rho W_r * \bar{r}(x)] = -v\rho(\partial_x W_r) * \bar{r}(x), \tag{14}$$

where we have used $\partial_s = -\partial_x$ (Eq. 7), and swapped the convolution and partial derivative. The above equation provides an insight that the translation generator can be converted into the derivative of recurrent connections in the network, i.e., $\partial_x W_r$. Therefore, it motivates us to consider a modified CAN dynamics whose recurrent connections have a new component ($\partial_x W_r$, Fig. 3A) [21, 29],

$$\tau\frac{\partial u(x,t)}{\partial t} = -u(x,t) + \rho\left(W_r - \tau v\partial_x W_r\right) * r(x,t). \tag{15}$$

Next we verify whether the modified CAN dynamics (Eq. 15) embeds translation operators, i.e., a spontaneously moving neural response along the stimulus manifold with speed $v$. We substitute the mean population response ($\bar{u}(x-s)$ and $\bar{r}(x-s)$, Eq. 10) into the above network dynamics, repeatedly use the stationary response that satisfies $\bar{u}(x-s) = \rho W_r * \bar{r}(x-s)$ to cancel terms,

$$\begin{aligned}\frac{\partial\bar{u}(x-s)}{\partial t} &= \tau^{-1}[-\bar{u}(x-s) + \rho W_r * \bar{r}(x-s)] - v\rho(\partial_x W_r) * \bar{r}(x-s), \\ &= -v\partial_x[\rho W_r * \bar{r}(x-s)] = v\hat{p}\cdot\bar{u}(x-s).\end{aligned} \tag{16}$$

Comparing Eqs. (16) and (13), we see the stationary state in the modified CAN spontaneously translates along the stimulus manifold in a way consistent with the action of translation operators (Eq. 13). Therefore the recurrent connection component $\partial_x W_r$ represents the translation generator, and its magnitude determines the moving speed $v$. Numerical simulations of the network model also confirm our theoretical derivations (Fig. 3B-C), where the actual moving speed is well predicted by the theory using the magnitude of recurrent connection $\partial_x W_r$.

When the network response deviates from the stereotypical Gaussian profile (Eq. 10), the recurrent connection component $W_r$ will remove distortions because the eigenvalues of all perturbations perpendicular to the continuous stimulus manifold are smaller than 1 (Eq. 12a - 12c). Once the population response settles into a smooth Gaussian profile, the network response translates along the continuous manifold with the desired speed (see details in SI. Sec. 4).

## 4.2 Path integration in the modified CAN dynamics

The brain accumulates the effects of a series of translation operators, and the combined translation effects are not dependent on their action orders (Eq. 3b), i.e., translations are commutative. The translation accumulation that is independent of action orders corresponds to the path integration in the neuroscience research [23, 29, 30], where the animals' navigation system integrates their walking path to update their walking distance. The path integration can be achieved in the modified CAN dynamics (Eq. 15). To illustrate this, we compute the translation distance in the modified CAN dynamics (Eq. 16) by using the chain rule, $\frac{\partial \bar{u}(x-s)}{\partial t} = \frac{\partial \bar{u}(x-s)}{\partial s} \frac{\partial s}{\partial t} = -\frac{\partial \bar{u}(x-s)}{\partial x} v(t)$, and substitute it back to Eq. (16). Finally, the translation distance is derived as

$$ds = v(t)dt, \tag{17}$$

where $ds$ is the translation distance along the continuous stimulus manifold in the CAN in a time window lasting $dt$. As an example, we consider a constant moving speed $v_1$ lasts for time duration $t_1$, followed by another constant speed $v_2$ lasting for duration $t_2$, and then we swap the order of two speeds (Fig. 3D, top). The moving distance in the two cases can be calculated as the same,

$$|s| = \int_0^{t_1} v_1 dt + \int_{t_1}^{t_1+t_2} v_2 dt = \int_0^{t_2} v_2 dt + \int_{t_2}^{t_2+t_1} v_1 dt = v_1 t_1 + v_2 t_2, \tag{18}$$

which shows the translation in the modified CAN is commutative, i.e., the network achieves path integration. This is also confirmed by numerical simulations (Fig. 3D, bottom).

# 5 A biologically plausible equivariant recurrent circuit

Although in the modified CAN dynamics the recurrent connection component $\partial_x W_r$ (Eq. 15) mathematically represents the translation generator and translates network responses, the connection $\partial_x W_r$ is not biologically plausible due to two reasons. First, the magnitude of the connection component $\partial_x W_r$ needs to be multiplicatively modulated by the speed $v$ (Eq. 15), but multiplicative modulation of synaptic weights is challenging in neural systems. Second, with Gaussian-profile connections (Eq. 9), the derivative $\partial_x W_r$ means that a neuron has both excitatory and inhibitory synapses, which disobeys Dale's law that a neuron can be either excitatory or inhibitory but cannot be both. Therefore, we propose a new neural circuit model to represent the translation operator that is consistent with the biological neural circuitry. Mathematically, we will replace the term of multiplicative synaptic weight modulation (Eq. 15, last term) by a biologically plausible input term that is originated from separate neuronal populations whose responses represent moving speed $v$.

The new circuit model is inspired by the Drosophila's compass system which consists of three neuronal populations (Fig. 4B, e.g., [31, 32]). There is a ring of heading-direction (stimulus $s$) neurons whose dynamics is governed by a CAN (Fig. 4B, E-PG neurons in blue disk, [33]). Besides, there are two populations of speed neurons (Fig. 4B, left and right P-EN neurons) whose firing rates encode the speed (translation distance $a$ in a unit time). Moreover, the feedback projections from speed neurons to heading-direction neurons are either clockwise or anti-clockwise *shifted* and they induce translations of heading-direction neurons' responses. Therefore, our new recurrent circuit model has three populations of neurons (Fig. 4A): the stimulus $s$ neurons which is modeled as a CAN as before (Eqs. 8a-8b), and two populations of speed neurons which are supposed to encode the positive and negative speed respectively. The network dynamics is governed by [31, 34–36],

$$\tau \frac{\partial u(x,t)}{\partial t} = -u(x,t) + \rho W_r * r(x,t) + \rho \sum_{m=\pm} W_m * r_m(x,t), \tag{19a}$$

$$\tau \frac{\partial u_\pm(x,t)}{\partial t} = -u_\pm(x,t) + w_{vs} r(x,t), \quad r_\pm(x,t) = [(g_v \pm v)u_\pm(x,t)]_+. \tag{19b}$$

Eq. (19a) is the CAN dynamics of stimulus neurons. $u_\pm(x,t)$ ($r_\pm(x,t)$) are the synaptic inputs (firing rate) of speed neurons which represent positive and negative speeds respectively and are driven

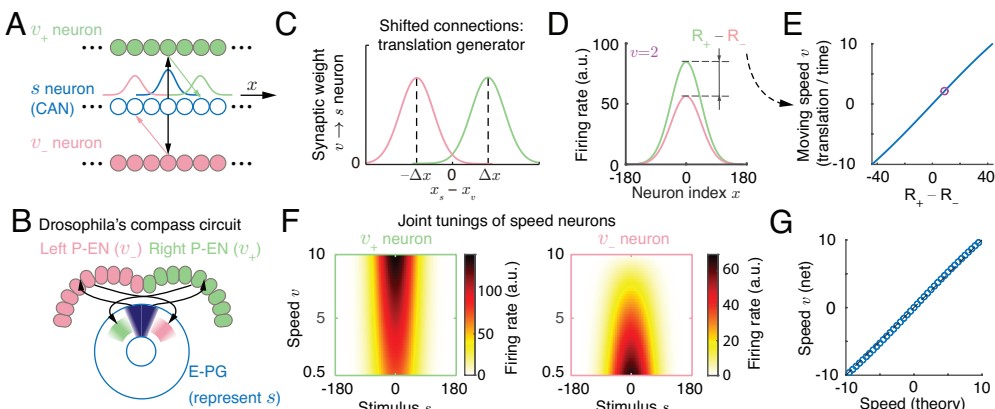

Figure 4: The network model with disentangled representations of stimulus $s$ and speed $v$ (amount of translation). (A-B) The proposed network structure (A) is similar to the Drosophila's internal compass (B). The three bumps shown in the middle represent the recurrent inputs from corresponding color coded neural populations. (C) The connection strength from speed neurons to stimulus neurons are shifted by $\Delta x$ which represents the translation generator. (D-E) The difference of firing rate between two populations of speed neurons (D) determines the moving speed of population activities of stimulus neurons (E). (F) The joint tuning curves over stimulus and speed of two example neurons. (G) The proposed network implements the path integration with consistent speed as theory.

by the stimulus neuron $r(x, t)$ preferring stimulus $x$. Note that stimulus neurons receive feedback inputs from speed neurons (Eq. 19a, last term), which replaces recurrent inputs from biologically implausible recurrent weight $\partial_x W_r$ (Eq. 15, last term). Consistent with the Drosophila's compass [31, 32, 34], recurrent connections $W_\pm(x, x')$ from the speed neuron $x'$ to the stimulus neuron $x$ is shifted by $\mp \Delta x$ towards the purported moving direction encoded by speed neurons (Fig. 4C),

$$W_\pm(x, x') = w_{sv}(\sqrt{2\pi}\sigma)^{-1} \exp\left[-(x - x' \mp \Delta x)^2/2\sigma^2\right]. \tag{20}$$

Moreover, speed neurons' firing rates $r_\pm(x, t)$ are multiplicatively modulated by the speed $v$ (Eq. 19b). This operation is necessary to reproduce gain modulation of speed neurons' responses (speed invariant stimulus tuning width) in present rate-based model [31], and it achieves the same computation with multiplicative modulation of synaptic weights (Eq. 15, last term). $g_v$ is a constant determining the baseline activity of speed neurons. To emphasize the main computational mechanism, we simplify the circuit model (Eqs. 19a-19b) compared with the Drosophila's circuit. First, we ignored the interactions between speed neurons. Second, we assumed that a speed neuron is only driven by a stimulus neuron with the same preferred stimulus, which is exhibited by the scalar $w_{vs}$ (Eq. 19b).

## 5.1 Neural representation of translation operators

We investigate how the abstract translation operator $\hat{T}(a)$ is represented in the circuit model inspired by the Drosophila's compass. Similar to the theoretical analysis performed in Eqs. (15-16), we focus on the latent network dynamics along the dominant continuous stimulus manifold by ignoring the fluctuations along other directions. Then Eq. (19a) can be simplified into (see more details in SI. 5),

$$\frac{\partial}{\partial t}\bar{u}(x - s) \approx \frac{\rho w_{sv}}{\sqrt{2}\tau U}(R_+ - R_-)\left[\Delta x\, \hat{p} \cdot \bar{u}(x - s)\right], \tag{21}$$

where $R_+$ and $R_-$ are the peak firing rates of positive and negative speed neurons, and $U$ is the peak synaptic input received by stimulus neurons (Eq. 10). The shifted connections $W_\pm(x, x')$ from speed neurons to stimulus neurons introduce the translation generator, in that the magnitude of translation generator is proportional to the connection shift $\Delta x$ (Eq. 16). The excitatory $W_\pm(x, x')$ pulls the network response towards shifted directions (implementing the excitatory part of the translation generator), while the inhibitory part of the translation generator is achieved through the network effect of global inhibition (Eq. 8b). Moreover, the firing rate difference between two populations of speed neuron, $R_+ - R_-$, determines the translation speed (1D translation operator parameter). Overall, the fixed translation generator $\hat{p}$ is stored in the shifted connections from speed to stimulus

neurons, and the time-varying translation speed is represented by the firing rate difference between two speed neuron populations which can change in a short time scale.

Meanwhile, the speed neurons' dynamics (Eq. 19b) suggests $R_+ - R_- = [(g_v + v) - (g_v - v)] = 2vw_{vs}R$. To translate network responses with the desired speed, the coefficient on the right hand side of Eq. (21) should equal to the speed determining speed neurons' response (Eq. 19b), which requires,

$$\sqrt{2}\rho w_{sv}w_{vs}R\Delta x = \tau U. \tag{22}$$

We performed numerical simulation of the circuit model (Eqs. 19a-19b), whose parameters are set to satisfy Eq. (22). We firstly verified our theoretical prediction, i.e., the moving speed is proportional to the firing rate difference between two populations of speed neurons, i.e., $R_+ - R_-$ (Fig. 4D-E). Indeed, the actual moving speed of stimulus neurons' population responses is consistent with the required speed that determines the firing rate of speed neurons (Fig. 4G), which supports the validity of our theoretical analysis and indicates the accurate path integration in the proposed circuit model. The joint tuning curves of two example speed neurons with respect to the stimulus $s$ and moving speed $v$ are plotted in Fig. 4F, which are multiplicatively modulated by the speed, i.e., the speed invariant stimulus tuning width, and this is consistent with experimental observations [31].

## 6    Conclusion and Discussion

Equivariant representation is necessary for the brain and artificial perceptual systems to reliably represent the stimulus under transformations [2–4, 6, 7, 37]. The present study investigates how recurrent neural circuits implement equivariant stimulus representation and Lie group operator representation. Our theoretical analysis shows that the stimulus can be equivariantly represented in a homogeneous neural population code that collectively emerges from a CAN dynamics. The Lie translation group operators are represented by extra neuronal populations whose activities represent the translation speed (translation group parameter) and whose feedback connections to the CAN store the translation generator. Our study for the first time links the Lie group equivariance to a concrete, biologically plausible recurrent neural circuit. It gives us insight into understanding the neural coding principle and may inspire a new building block for equivariant representation in ML tasks.

The CAN is a canonical neural circuit model widely used in neuroscience [20, 22, 25, 38, 39]. Recurrent circuit models similar to the present model were proposed before (e.g., [29, 31, 34–36, 40]), whereas no previous studies theoretically linked the network dynamics to the equivariant stimulus representation, nor the Lie group operator representation. Besides, our theory also derives the condition that network responses translate with the desired speed (Eq. 22), which is testable by analyzing experimental data.

To highlight the basic mechanism of equivariant representation in a recurrent circuit, the proposed neural circuit model has some simplifications compared with neural circuits in the brain. First, we considered homogeneous neurons and ignored the randomness on recurrent connections and neural noises in real cortical circuits. Apparently, the neural noise and random connections will break the perfect structure of equivariant representation. It has been suggested that the CAN dynamics is resistant to input noise by removing the noise component that is perpendicular to the attractor manifold [24, 41], and then the equivariant representation can be approximately held. Another possibility is that the neural heterogeneity and neural noise can help the brain implement sampling-based Bayesian inference (e.g., [42–47]), where the instant response of the CAN and the activity of speed neurons represent the samples of heading direction and rotation speed, respectively (e.g., [48, 49]). Second, we analyzed a limiting case of infinite number of neurons in a recurrent circuit model, resulting in that the translation operator (Eq. 2) acts on a continuous function of neuronal population response. Although it brings huge benefit in theoretical analysis, the real neural circuit only has finite number of neurons, especially there are only dozens of E-PG neurons in Drosophila's heading direction circuit (Fig. 4A, [31, 32]). In the case of finite number of neurons, the recurrent circuit model can still have equivariant representation, however, the profile of population response and the convolution kernel will slightly deviate from Gaussian and the concrete profile can be obtained by numerical optimization, as suggested by a recent work [40].

The proposed neural circuit model can be extended to capture more characteristics in real neural circuits. For example, our model doesn't have the sensory input in order to emphasize an internal mechanism inducing the translation of neural representation. The sensory inputs that represent

instantaneous stimulus direction and the instantaneous velocity into Eqs. (19a and 19b) respectively, similar with in earlier results [31, 34]. Moreover, the neural circuit model considers Gaussian profile of population responses, while neuronal tunings sometimes have long tail with high kurtosis [50]. In principle, the recurrent connection profile can be varied to let the recurrent circuit self-consistently generate different tuning curves. In particular, the high kurtosis profile may be considered a scale-mixture of Gaussians, i.e., Gaussians with different scales, and then it can be generated by recurrent connections that is also consist of mixture of many Gaussian connection profiles

To elucidate the concepts clearly, the present study only investigates the 1D translation group while the proposed framework can be generalized to more complicated group transformations. For example, we may develop a recurrent circuit with a 2D translation group equivariant representation to explain the spatial location coding in place and grid cells (e.g., [30, 51, 52]). A recent study reported that the population activities of grid cells appear to reside in a 2D torus manifold [53], which is the 2D version of the ring that is being studied in the present study. Meanwhile the CAN based on 2D torus was also proposed in previous studies (e.g., [30, 48]). And therefore it is possible to generalize our current theoretical framework to the 2D domain to partially explain the properties of grid cells in future work. Another possibility is developing a (non-commutative) 3D rotation group equivariant recurrent circuits to explain the 3D rotation encoding in the brain. These studies form our future research.

## Acknowledgments

Y. W. is supported by NSF DMS-2015577. S. W. is supported by Science and Technology Innovation 2030-Brain Science and Brain-inspired Intelligence Project (No. 2021ZD0200204) and Beijing Academy of Artificial Intelligence.

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
