# Translation-equivariant Representation in Recurrent Networks with a Continuous Manifold of Attractors: Supplementary Information

**Wen-Hao Zhang**[1,2*]
wenhao.zhang@utsouthwestern.edu

**Ying Nian Wu**[3]
ywu@stat.ucla.edu

**Si Wu**[4]
siwu@pku.edu.cn

[1]Lyda Hill Department of Bioinformatics, UT Southwestern Medical Center.
[2]O'Donnell Brain Institute, UT Southwestern Medical Center.
[3]Department of Statistics, UCLA.
[4]School of Psychology and Cognitive Sciences, IDG/McGovern Institute for Brain Research, Center of Quantitative Biology, Peking-Tsinghua Center for Life Sciences, Peking University.

## Contents

---

*Corresponding author.

36th Conference on Neural Information Processing Systems (NeurIPS 2022).

# 1 The one-dimensional translation Lie group

We consider a 1D translation Lie group $\mathbb{T}$ which translates the 1D stimulus $s$. A group element $T(a) \in \mathbb{T}$ translates the stimulus $s$ to $s + a$, and mathematically it can be denoted as $T(a) \cdot s = s + a$ where $\cdot$ denotes a group action. Based on the requirement of equivariant representation (Eq. 1), a neural representation $\bar{u}(s)$ (a continuous function over $s$ and is defined on the $L^2$ Hilbert space) which is equivariant to the 1D translation group should satisfy (Fig. 1C),

$$\bar{u}\big[T(a) \cdot s\big] = \bar{u}(s + a) = \hat{T}(a) \cdot \bar{u}(s), \tag{S1}$$

where $\hat{T}(a)$ is the 1D translation operator inducing the translation of neural representation (Fig. 1C). It is worthy to distinguish that $\hat{T}(a)$ and $T(a)$ act on different spaces: $T(a)$ induces translation on the original stimulus $s$ space, while $\hat{T}(a)$ is the translation operator acting on the space of neural representation $\bar{u}(s)$. From the definition (Eq. 2) we could directly derive the properties of 1D translation operators $\hat{T}(a)$ [1],

$$\hat{T}(0) = 1, \tag{S2a}$$
$$\hat{T}(a)\hat{T}(b) = \hat{T}(a + b) = \hat{T}(b)\hat{T}(a), \tag{S2b}$$
$$\hat{T}(a)^{-1} = \hat{T}(-a), \tag{S2c}$$

whose intuitive explanations can be found at the text right below Eq. (3c) in the main text.

Since the translation is continuous, the amount of translation can be made infinitesimally small. Consider an infinitesimal translation $\delta a \to 0$ on the stimulus $s$, then the corresponding change of neural representation is,

$$\hat{T}(\delta a) \cdot \bar{u}(s) = \bar{u}(s + \delta a),$$

Taking a first order Taylor expansion of above equation,

$$\begin{aligned}\hat{T}(\delta a) \cdot \bar{u}(s) &\approx \bar{u}(s) + \delta a \partial_s \bar{u}(s), \\ &= (1 + \delta a \partial_s) \cdot \bar{u}(s), \\ &= (1 + \delta a \hat{p}) \cdot \bar{u}(s), \end{aligned} \tag{S3}$$

where $\partial_s = \partial/\partial s$ denotes the derivative over $s$. Also in Eq. (S3) we define

$$\hat{p} \equiv \partial_s, \tag{S4}$$

as the translation *generator*. The generator and characterizes the tangential direction of translation in the group space, and forms a basis of the Lie *algebra*. $\hat{p}$ is usually regarded as the momentum operator in physics [1].

Eq. (S2b) suggests a large translation can be decomposed as a composition of many infinitesimal translations,

$$\hat{T}(a) = \underbrace{\hat{T}\left(\frac{a}{N}\right) \cdots \hat{T}\left(\frac{a}{N}\right)}_{N} \equiv \left[\hat{T}\left(\frac{a}{N}\right)\right]^N. \tag{S5}$$

Considering the large limit of $N$ and utilizing Eq. (S3), the above equation can be converted into

$$\begin{aligned}\hat{T}(a) &= \lim_{N \to \infty} \left[\hat{T}\left(\frac{a}{N}\right)\right]^N, \\ &\approx \lim_{N \to \infty} \left(1 + \frac{a}{N}\hat{p}\right)^N, \\ &= \exp(a\hat{p}). \end{aligned} \tag{S6}$$

It is clear to see the translation operator $\hat{T}$ is an exponential map of the translation generator $\hat{p}$. Differentiating the above equation we can derive a differential form of a translation operator,

$$\frac{d\hat{T}(a)}{da} = \hat{p} \cdot \exp(a\hat{p}) = \hat{p} \cdot \hat{T}(a). \tag{S7}$$

The exponential map from the translation generator $\hat{p}$ to the translation operator $\hat{T}(a)$ implies that computing a translation operator can be well implemented by a recurrent neural dynamics, because the differential form (Eq. S2b) is well consistent with a differential neural dynamics. This motivates us to propose a recurrent neural dynamics to compute translation group operators.

## 2 Stationary population responses of the CAN

We present the math of verifying the Gaussian ansatz of the stationary population responses in the CAN (Eq. 10). If the Gaussian ansatz was correct, based on Eq. (8a) they should satisfy that

$$\bar{u}(x-s) = \rho \int \mathsf{W_r}(x-x') \cdot \bar{r}(x'-s)dx', \tag{S8}$$

Substituting the Gaussian ansatz $\bar{u}(x-s)$ and $\bar{r}(x-s)$ respectively into the left and right hand sides of above equation,

$$\mathsf{U}e^{-(x-s)^2/4\sigma^2} = \rho \int \mathbf{W_r}(x-x') \cdot \bar{r}(x'-s)dx',$$

$$= \frac{\rho \mathsf{w_r} \mathsf{R}}{\sqrt{2\pi}a} \int e^{-(x-x')^2/2\sigma^2 - (x'-s)^2/2\sigma^2} dx',$$

$$= \frac{\rho \mathsf{w_r} \mathsf{R}}{\sqrt{2}} e^{-(x-s)^2/4\sigma^2}.$$

We see the left and right hand sides in above equations contain the same Gaussian terms with the same position $s$ and width $\sigma$. Equating the magnitude on the two sides in above equation we have,

$$\mathsf{U} = \frac{\rho \mathsf{w_r} \mathsf{R}}{\sqrt{2}}. \tag{S9}$$

Meanwhile, substituting the Gaussian ansatz (Eq. 10) into the activation function of the network dynamics (Eq. 8b), it reads as

$$\mathsf{R} = \frac{\mathsf{U}^2}{1 + \sqrt{2\pi}k\rho\sigma\mathsf{U}^2}. \tag{S10}$$

Combining Eqs. (S9 and S10), we get a quadratic equation of $\mathsf{U}$,

$$2\sqrt{\pi}k\rho\sigma\mathsf{U}^2 - \rho\mathsf{w_r}\mathsf{U} + \sqrt{2} = 0.$$

It can be computed that when the recurrent weight $\mathsf{w_r}$ is larger than a critical value

$$\mathsf{w_c} = 2\sqrt{2}(2\pi)^{1/4}\sqrt{k\sigma/\rho}, \tag{S11}$$

the CAN exists a stable non-zero population response whose magnitude is

$$\mathsf{U} = \frac{\mathsf{w_r}(1 + \sqrt{1 - \mathsf{w_c}^2/\mathsf{w_r}^2})}{4\sqrt{\pi}k\sigma}, \mathsf{R} = \frac{1 + \sqrt{1 - \mathsf{w_c}^2/\mathsf{w_r}^2}}{2\sqrt{2\pi}\rho k\sigma}. \tag{S12}$$

## 3 The perturbative dynamics of the CAN

We performed perturbative analysis to analyze the stability of the CAN dynamics. We add a small perturbation $\delta u(x, t)$ on the stationary state $\bar{u}(x-s)$, i.e., $u(x, t) = \bar{u}(x-s) + \delta u(x, t)$, and substitute $u(x, t)$ into the CAN dynamics (Eq. 8a). And then the dynamics of perturbation $\delta u(x, t)$ can be derived as [2],

$$\tau\frac{\partial}{\partial t}\delta u(x, t) = -\delta u(x, t) + \int K(x, x'|s)\delta u(x', t)dx', \tag{S13}$$

where the interaction kernel is

$$K(x, x'|s) = \rho \int \mathsf{W_r}(x - x'')\frac{\partial \bar{r}(x'' - s)}{\partial \bar{u}(x' - s)}dx'',$$

$$= \frac{2\rho\bar{u}(x' - s)}{D}\left[\mathsf{W_r}(x, x') - k\rho \int \mathsf{W_r}(x, x'')\bar{r}(x'' - s)dx''\right],$$

$$= \frac{2}{\sigma\sqrt{\pi}}\exp\left[-\frac{(x - x')^2}{2\sigma^2} - \frac{(x' - s)^2}{4\sigma^2}\right],$$

$$- \frac{1 + \sqrt{1 - \mathsf{w_c}^2/\mathsf{w_r}^2}}{\sqrt{2\pi}\sigma}\exp\left[-\frac{(x - s)^2}{4\sigma^2} - \frac{(x' - s)^2}{4\sigma^2}\right]. \tag{S14}$$

The $D$ in above equation denotes the magnitude in the divisive normalization pool (the denominator of divisive normalization in Eq. 8b)

$$D = 1 + k\rho \int \bar{u}(x')^2 dx' = \rho \mathsf{w_r} \mathsf{U}/\sqrt{2},$$

where $\mathsf{U}$ denotes the magnitude of population synaptic inputs (Eq. 10).

### 3.1 Eigenspetrum of the interaction kernel

A previous theoretical study of the continuous attractor dynamics [2] suggested that the wave functions of quantum harmonic oscillators can be used as basis functions of the perturbed CAN dynamics, and then eigenfunctions of the interaction kernel can be expressed as linear combinations of at most two of these basis functions. The wave functions of quantum harmonic oscillators is,

$$v_n(x|s) = \frac{(-1)^n (\sqrt{2}\sigma)^{n-1/2}}{\sqrt{\pi^{1/2} n! 2^n}} \exp\left[\frac{(x-s)^2}{4\sigma^2}\right] \left(\frac{d}{dx}\right)^n \exp\left[-\frac{(x-s)^2}{2\sigma^2}\right].$$

For illustration, we write the explicit expression of wave functions with first and second orders,

$$v_0(x|s) = \frac{\bar{u}(x-s)}{\mathsf{U}\sqrt{(2\pi)^{1/2}\sigma}} \propto \bar{u}(x-s),$$

$$v_1(x|s) = \frac{2\sigma}{\mathsf{U}\sqrt{(2\pi)^{1/2}\sigma}} \frac{\partial \bar{u}(x-s)}{\partial s} \propto \partial_s \bar{u}(x-s).$$

We see $v_0(x|s)$ and $v_1(x|s)$ corresponds to respectively the change of the magnitude of population responses, and the translation of population responses along the stimulus manifold. In particular, $v_1(x|s)$ is proportional to the translation generator (Eq. S4).

Treating the interaction kernel $K(x, x'|s)$ as an operator, we next compute its eigenvalues and eigenfunctions. Since $K(x, x'|s)$ is not symmetric, we need to distinguish its left and right eigenfunctions. In the perturbed network dynamics, $K(x, x'|s)$ acts on the perturbation $\delta u(x, t)$ from the left side (Eq. S13), hence we compute the right eigenfunctions of $K(x, x'|s)$ [2]. Below we list few dominant eigenfunctions and their corresponding eigenvalues,

$$\lambda_1 = 1, \qquad\qquad f_1(x|s) = v_1(x|s), \tag{S15a}$$

$$\lambda_2 = 1 - \sqrt{1 - \mathsf{w_c}^2/\mathsf{w_r}^2}, \quad f_2(x|s) = v_0(x|s), \tag{S15b}$$

$$\lambda_3 = 1/2, \qquad\qquad f_3(x|s) \propto 2^{-1/2} v_0(x|s) + (1 - 2\sqrt{1 - \mathsf{w_c}^2/\mathsf{w_r}^2}) v_2(x|s). \tag{S15c}$$

$$\lambda_4 = 2^{-2}, \qquad\qquad f_4(x|s) = v_1(x|s) + \sqrt{6} v_3(x|s), \tag{S15d}$$

$$\lambda_n = 2^{2-n} \ (n \geq 3), \qquad f_5(x|s) = \cdots. \tag{S15e}$$

To keep notations concise, some eigenfunctions in above equations are not normalized to have a unit L2 norm.

## 4  Translation operator in the derivative of recurrent connections

We performed theoretical analysis to verify whether the modified CAN dynamics with a derivative of recurrent connection component (Eq. 15) could translate neural responses along the stimulus manifold in a way consistent with the action of translation operator as shown in Eq. (13). In theory, we consider the instantaneous state $u(x, t)$ is perturbed around the attractor state $\bar{u}(x-s)$ in the original CAN dynamics (Eqs. 8a- 8b),

$$u(x, t) = \bar{u}(x-s) + \delta u(x, t).$$

Substituting above equation into the modified CAN dynamics (Eq. 15) and utilizing the interaction kernel (Eq. S14),

$$\tau \frac{\partial}{\partial t} [\bar{u}(x-s) + \delta u(x, t)] = - [\bar{u}(x-s) + \delta u(x, t)],$$

$$+ \rho \mathsf{W_r} * \bar{r}(x-s) + \int K(x, x'|s)\delta u(x', t)dx', \tag{S16}$$

$$- \tau v \partial_x \bar{u}(x-s) - \tau v \partial_x \left[\int K(x, x'|s)\delta u(x', t)dx'\right].$$

Using the condition that $\bar{u}(x - s) = \rho \mathsf{W_r} * \bar{r}(x - s)$, the Eq. (S16) can be simplified into

$$\tau \frac{\partial}{\partial t} [\bar{u}(x - s) + \delta u(x, t)] = - \delta u(x, t) + \int K(x, x'|s) \delta u(x', t) dx',$$
$$- \tau v \partial_x \bar{u}(x - s) - \tau v \partial_x \left[ \int K(x, x'|s) \delta u(x', t) dx' \right],$$

To simplify above equation further, we express the perturbation $\delta u(x, t)$ as a linear combination of eigenfunctions of $K(x, x'|s)$,

$$\delta u(x, t) = \sum_n a_n f_n(x|s),$$

and therefore the action of kernel $K(x, x'|s)$ on $\delta u(x, t)$ becomes

$$\int K(x, x'|s) \delta u(x', t) dx' = \sum_n \lambda_n a_n f_n(x|s).$$

Therefore, Eq. (S16) can be simplified as,

$$\tau \frac{\partial}{\partial t} \bar{u}(x - s) + \tau \sum_n \frac{da_n}{dt} f_n(x|s) = - \tau v \partial_x \bar{u}(x - s) + \sum_n (\lambda_n - 1) a_n f_n(x|s),$$
$$- \tau v \partial_x \left[ \sum_n \lambda_n a_n f_n(x|s) \right]. \tag{S17}$$

Comparing Eq. (S17) with the required dynamics derived from translation group (Eq. 13), we see if all $a_n$ disappear the Eq. (S17) will become exactly the same as Eq. (13). Thus it motivates us to analyze the dynamics of $a_n$ that characterizes the magnitude of perturbations.

Since the eigenfunctions $f_n(x|s)$ form a orthogonal and complete basis set of the perturbed dynamics (Eq. S15e), we could project the Eq. (S17) onto each eigenfunction $f_n(x|s)$ to compute the temporal evolution of $a_n$. The projection corresponds to compute the inner product between the network state $u(x, t)$ and the $f_n(x|s)$,

$$\langle u(x), f_n(x) \rangle = \int u(x) f_n(x) dx. \tag{S18}$$

Using the orthogonality between eigenfunctions, the projection of Eq. (S17) on $f_n(x|s)$ can be computed as

$$\tau \frac{\partial}{\partial t} \langle \bar{u}(x - s), f_n(x|s) \rangle + \tau \frac{da_n}{dt} = - \tau v \partial_x \langle \bar{u}(x - s), f_n(x|s) \rangle + (\lambda_n - 1) a_n - \tau v \partial_x (\lambda_n a_n).$$

Moreover, the stationary state $\bar{u}(x - s) \propto f_2(x|s)$ (Eq. S15e), and therefore

$$\langle \bar{u}(x - s), f_n(x|s) \rangle \propto \delta_{n,2} \langle f_2(x|s), f_2(x|s) \rangle \equiv \delta_{n,2} |f_2(x|s)|^2,$$

where $\delta_{n,2}$ is a Kronecker delta function. Since the L2 norm $|f_2(x|s)|^2$ is a constant, its derivatives will be zero, i.e.,

$$\frac{\partial}{\partial t} \langle \bar{u}(x - s), f_n(x|s) \rangle = \frac{\partial}{\partial t} |f_2(x|s)|^2 = 0,$$
$$\frac{\partial}{\partial x} \langle \bar{u}(x - s), f_n(x|s) \rangle = \frac{\partial}{\partial x} |f_2(x|s)|^2 = 0.$$

Finally the projection of Eq. (S17) onto $f_n(x|s)$ can be calculated as,

$$\tau \frac{da_n}{dt} = (\lambda_n - 1) a_n + \tau v \partial_x (\lambda_n a_n),$$
$$= (\lambda_n - 1) a_n, \tag{S19}$$

where we use that $\partial_x (\lambda_n a_n) = 0$. The eigenvalues $\lambda_n$ for $n \geq 2$ are smaller than 1, indicating the coefficients $a_n$ ($n \geq 2$) will eventually decay to zero and then they can be ignored. Meanwhile,

$\lambda_1 = 1$ indicates the coefficient $a_1$ will remain constant over time. Assuming $a_1 = 0$, all terms containing $a_n$ in Eq. (S17) will vanish in the end,

$$\frac{\partial}{\partial t}\bar{u}(x-s) \approx -v\partial_x\bar{u}(x-s) = v\hat{p}\cdot\bar{u}(x-s). \tag{S20}$$

Comparing Eq. (S20) and the Eq. (13), we see the modified CAN dynamics with a derivative of recurrent connection component (Eq. 15) will firstly remove the pertubations of network activities to get smooth Gaussian-profile network responses, and then the stationary network responses can self-consistently translate with speed $v$ along the stimulus manifold.

## 5 The network model inspired from Drosophila's compass circuit

### 5.1 Stationary network responses

Similar with the analysis in the CAN, we propose the following Gaussian ansatz of network's stationary responses (Eqs. 19a and 19b),

$$\begin{aligned}
\bar{u}(x-s) &= \mathsf{U}e^{-(x-s)^2/4\sigma^2}, \quad \bar{r}(x-s) = \mathsf{U}e^{-(x-s)^2/2\sigma^2}, \\
\bar{u}_\pm(x-s) &= \mathsf{U}_\pm e^{-(x-s)^2/2\sigma^2}, \bar{r}_\pm(x-s) = \mathsf{R}_\pm e^{-(x-s)^2/4\sigma^2}.
\end{aligned} \tag{S21}$$

For simplicity, we assume the speed neurons' responses $\bar{u}_\pm(x-s)$ and $\bar{r}_\pm(x-s)$ have the same position $s$ on the stimulus manifold with the stimulus neurons' responses, i.e., $\bar{u}(x-s)$ and $\bar{r}(x-s)$. This simplification is equivalent to assume that the transmission delay from stimulus neurons to speed neurons and the time constant of speed neurons are small enough. Substituting the above Gaussian ansatz into the network dynamics of stimulus neurons (Eq. 19a),

$$\begin{aligned}
\frac{\tau\mathsf{U}}{2\sigma^2}\frac{ds}{dt}(x-s)e^{-(x-s)^2/4\sigma^2} &= -\mathsf{U}e^{-(x-s)^2/4\sigma^2} + \frac{\rho\mathsf{w_r}\mathsf{R}}{\sqrt{2}}e^{-(x-s)^2/4\sigma^2}, \\
&+ \frac{\rho\mathsf{w}_{sv}}{\sqrt{2}}\sum_{m=\pm}\mathsf{R}_m e^{-(x-s-m\Delta x)^2/4\sigma^2}.
\end{aligned} \tag{S22}$$

### 5.2 Translation in the network model

In order to study whether and how feedback inputs from speed neurons to stimulus neurons (the last term in above equation) induce translations on stimulus neurons' responses, we project Eq. (S22) onto the eigenfunction $f_1(x|s)$ corresponding to the translation along the continuous stimulus manifold. The projection is computing the inner product between the network dynamics (Eq. S22) and $f_1(x|s)$ (Eq. S18). Denoting the $f_1(x|s) = Z^{-1}(x-s)e^{-(x-s)^2/4\sigma^2}$ with $Z$ a normalization factor, we list the major calculations of the projection in the text below.

$$\begin{aligned}
\text{LHS} &= \left\langle \frac{\tau\mathsf{U}}{2\sigma^2}\frac{ds}{dt}(x-s)e^{-(x-s)^2/4\sigma^2}, f_1(x|s) \right\rangle, \\
&= \frac{\tau\mathsf{U}}{2\sigma^2}\frac{ds}{dt}\frac{1}{Z}\int(x-s)^2 e^{-(x-s)^2/2\sigma^2}dx, \\
&= \frac{\tau\mathsf{U}}{2Z}\frac{ds}{dt}\sqrt{2\pi}\sigma,
\end{aligned}$$

The projection of the first two terms on the RHS of Eq. (S22) on $f_1(x|s)$ would be zero, because

$$\left\langle e^{-(x-s)^2/4\sigma^2}, Z^{-1}(x-s)e^{-(x-s)^2/4\sigma^2} \right\rangle \propto \int(x-s)e^{-(x-s)^2/2\sigma^2}dx = 0.$$

At last, the projection of the last term of Eq. (S22) on $f_1(x|s)$ is

$$\frac{\rho \mathsf{w}_{sv}}{\sqrt{2}Z} \sum_{m=\pm} \mathsf{R}_m \left\langle e^{-(x-s-m\Delta x)^2/4\sigma^2}, (x-s)e^{-(x-s)^2/4\sigma^2} \right\rangle,$$

$$= \frac{\rho \mathsf{w}_{sv}}{\sqrt{2}Z} \sum_{m=\pm} \mathsf{R}_m \int [(x-s-m\Delta x/2) + m\Delta x/2] \exp\left[-\frac{(x-s-m\Delta x/2)^2}{2\sigma^2} - \frac{\Delta x^2}{8\sigma^2}\right] dx,$$

$$= \frac{\rho \mathsf{w}_{sv}\Delta x}{2\sqrt{2}Z} \sum_{m=\pm} m\mathsf{R}_m \exp\left(-\frac{\Delta x^2}{8\sigma^2}\right) \int \exp\left[-\frac{(x-s-m\Delta x/2)^2}{2\sigma^2}\right] dx,$$

$$= \frac{\rho \mathsf{w}_{sv}\Delta x}{2\sqrt{2}Z} \sqrt{2\pi}\sigma \exp\left(-\frac{\Delta x^2}{8\sigma^2}\right) \sum_{m=\pm} m\mathsf{R}_m.$$

Combining all above calculations, the projection of Eq. (S22) onto the stimulus translation direction can be eventually computed as

$$\begin{aligned}
\frac{ds}{dt} &= \frac{\rho \mathsf{w}_{sv}\Delta x}{\sqrt{2}\tau\mathsf{U}} \sum_{m=\pm} m\mathsf{R}_m e^{-\Delta x^2/8a^2}, \\
&= \frac{\rho \mathsf{w}_{sv}\Delta x}{\sqrt{2}\tau\mathsf{U}} (\mathsf{R}_+ - \mathsf{R}_-)e^{-\Delta x^2/8a^2}.
\end{aligned} \tag{S23}$$

Suppose the amount of connection shift $\Delta x$ is small enough compared with the connection width $a$, i.e., $(\Delta x)^2 \ll 8a^2$, the exponential terms in above equation can be ignored for simplicity. Reorganize the terms in above equation,

$$\frac{ds}{dt} \approx \frac{\rho \mathsf{w}_{sv}}{\sqrt{2}\tau\mathsf{U}} (\mathsf{R}_+ - \mathsf{R}_-)\Delta x. \tag{S24}$$

In the stationary state, the network only translates along the stimulus space since the perturbations along other directions are removed by network dynamics (Eq. S19), and then we can derive the Eq. (21) in the main text,

$$\begin{aligned}
\frac{\partial \bar{u}(x-s)}{\partial t} &= \frac{\partial \bar{u}(x-s)}{\partial s}\frac{\partial s}{\partial t}, \\
&= \frac{\partial s}{\partial t}[\hat{p} \cdot \bar{u}(x-s)], \\
&= \frac{\rho \mathsf{w}_{sv}}{\sqrt{2}\tau\mathsf{U}} (\mathsf{R}_+ - \mathsf{R}_-)\Delta x[\hat{p} \cdot \bar{u}(x-s)].
\end{aligned} \tag{S25}$$

### 5.3 Translating with desired speed

The actual translation speed $v$ of network responses should be the same as the speed which determines the firing rate of speed neurons (Eq. 19b). We calculate how the network model satisfies this requirement. Combining the Gaussian ansatz (Eq. S21) and the activation function of speed neurons (Eq. 19b), and supposing the speed $v$ is smaller than the baseline activity $g_v$,

$$\mathsf{R}_+ - \mathsf{R}_- = [(g_v + v) - (g_v - v)]\mathsf{w}_{vs}\mathsf{R} = 2v\mathsf{w}_{vs}\mathsf{R}.$$

Substituting above equation into Eq. (S24),

$$\frac{ds}{dt} = \frac{\sqrt{2}\rho \mathsf{w}_{sv}\mathsf{w}_{vs}\mathsf{R}\Delta x}{\tau\mathsf{U}}v.$$

In order to translate the stimulus neurons' population response with the desired speed $v$, the coefficient on the right hand side of above equation should be one,

$$\sqrt{2}\rho \mathsf{w}_{sv}\mathsf{w}_{vs}\mathsf{R}\Delta x = \tau\mathsf{U}. \tag{S26}$$

Now we arrive the Eq. (22) in the main text.

# 6 Network simulation details

The typical set of network parameters can be found in Table 1. In the text below we briefly explain the reasoning of network parameter setting. We simulated a continuous attractor network that is consist of $N = 180$ neurons which are uniformly distributed in the space of $s$. To avoid boundary effect, we consider $s$ is a periodic variable in the network simulation and is in the range of $(-180°, 180°]$. The periodic stimulus $s$ doesn't affect substantially our theoretical results from the 1d translation group which acts on an infinite region, as long as the connection width, i.e., $\sigma$ in Eq. (9), is much smaller than the width of the stimulus space. In this setting, the neuronal density $\rho = N/360° = 0.5/\circ$. The connection width $\sigma = 40°$, which is consistent with experimental data, and also make the width of population responses smaller than the range of periodic stimulus space. The synaptic time constant $\tau$ is rescaled to 1 as a dimensionless number. And the global connection strength $k = 5 \times 10^{-4}$ (Eq. 8b), which makes the peak firing rate R of the network (Eq. 10) saturate around 50Hz, consistent with typical experimental observations. The network is simulated by using Euler method with time step $\Delta t = 0.01\tau$. The code was written in Matlab R2022b and was simulated on MacBookPro laptop which has a 10-core M1 CPU and 32GB RAM.

To scale the connection strength in the network model, we set the connection strength relative to the critical strength, $w_c$, under which a CAN can hold a persistent (non-zero) population response by itself. Based on Eq. (S11) and the actual parameters in the network, it can be calculated that

$$w_c = 2\sqrt{2}(2\pi)^{1/4}\sqrt{k\sigma/\rho} \approx 0.896. \tag{S27}$$

In the network simulation, the instantaneous stimulus representation $s_t$ is linearly read out from the stimulus neurons' response in the CAN, $\mathbf{r}(x,t)$ by using the population vector (Eq. [3]), i.e.,

$$s_t = \text{Angle}\Big[\sum_j r(x_j,t)e^{ix_j}\Big]. \tag{S28}$$

where $i = \sqrt{-1}$ is the pure imaginary number.

| Symbol | Description | Typical values (range) |
|---|---|---|
| $N$ | Number of (excitatory) neurons | 180 |
| $\rho$ | Neuronal density in the stimulus space | 0.5 |
| $\sigma$ | Tuning width | $40°$ |
| $k$ | Global inhibition strength | $5 \times 10^{-4}$ |
| $\tau$ | Synaptic decaying time constant | 1 (dimensionless) |
| $dt$ | Time step in numerical simulation | $0.01\tau$ |
| $w_r$ | The peak recurrent weight in the CAN | $w_c$ |
| $w_r$ | The critical recurrent weight of non-zero sustained response | 0.896 (Eq. S27) |
| $w_{vs}$ | The recurrent weight from stimulus to speed neurons | 0.2 |
| $w_{sv}$ | The recurrent weight from speed to stimulus neurons | 1 |
| $\Delta x$ | The connection shift from speed neurons to stimulus neurons | $22°$ |
| $g_v$ | The baseline activity of speed neurons | 10 |

Table 1: Typical parameters of the network model