# OpenReview forum: "Translation-equivariant Representation in Recurrent Networks with a Continuous Manifold of Attractors"
_NeurIPS.cc/2022/Conference — NeurIPS 2022 Accept_

### Official Review · Reviewer_fomB · 2022-07-09

**Rating:** 7
**Confidence:** 2
**Soundness:** 3 good
**Presentation:** 3 good
**Contribution:** 3 good

**Summary:**

The authors propose a recurrent neural circuit model with a continuous attractor state that models the heading direction system in the Drosophila and includes a circuit mechanism for translating the heading direction at arbitrary speeds. Moreover, their model is closely related to the Drosophila heading direction circuit.

**Questions:**

- Line 37: The sentence "For example, ..." needs a citation.
- Line 96: Why is eq 6 the mean response of all neurons? Are you assuming infinitely many neurons? Is there a citation justifying this? Am I missing something obvious?
- Eq (7): What does $\overline u(s)\equiv\overline{u}(x-s)$ mean? This doesn't appear to be true based on the definition of $\overline{u}(x-s)$.
- Sec 3.1: Is there no external input to the system? Is it easy to include? How would that affect the system?
- Line 131: What is the definition of "free parameter"?
- Eq (13): Can I interpret this effectively as a restatement of the fact that $\hat T(a)=\exp(a\hat p)$?
- To what extent are eqs (16) and (19) equivalent? The authors discuss the relationship, but its not clear to me what the precise relationship is.

Minor:
- Line 20: reflects -> reflect
- Line 43: whereas -> however?
- Eq (6): I find the notation $\cup_j$ confusing. It looks like you're taking the union over $j$.
- Eq (10): The definition of $\overline{u}$ is different here than in eq. (7)

**Limitations:**

- Eq (19ab): Can external stimulus inputs be included in this model? It seems like those would be reconciled with the speed inputs via some sort of Kalman filter. Is the speed $v$ assumed to be an external stimulus that feeds into all of the speed neurons? Presumably the speed is computed in part from the CAN responses? I'm not suggesting that this circuit should include these computations, rather I'm trying to understand to what degree this model is flexible enough to include these considerations.

**Strengths And Weaknesses:**

Originality: As someone who is not an expert in the field of continuous attractor networks, it is difficult for me to assess the originality of this work in part because it was not clear what exactly were the authors' contributions. My impression is that the main contributions were stated in sections 4 and 5, but even within those sections it is not clear which parts are novel (e.g., equation 19 appears to be known). While they include numerous references and the second paragraph of the conclusion discusses on the novelty of this work, it would still be useful (for me) if the authors further delineate what is their contribution versus what has already been established in the literature.

Quality: As far as I can tell, the paper is technically sound.

Clarity: I did not find the presentation particularly clear, but this could be due to my lack of expertise in the area. I've listed clarifying questions below.

Significance: The paper address an interesting problem in neuroscience; however, it is difficult for me to asses exactly what the authors' contributions are versus what was already known. I readily acknowledge this may be my own lack of expertise, but I would like the authors to clarify this for me.

---

> ### Author Response · Authors · 2022-07-31
> **Reply to reviewer's comments on weakness and limitations (1st part)**
>
> Thank you for the insightful review and interesting questions. We shall follow your suggestions to improve the presentation of our paper.
>
> ### Novelty and main contributions
>
> The novelty and main contributions of this paper are that it provides overarching connections between Lie group, continuous attractor network (CAN), and the drosophila’s neural circuit, and demonstrates for the first time how recurrent neural circuitry in the brain implements equivariant stimulus presentation. Here, we present a new perspective from the Lie group theory to understand CAN as a way for the brain to implement equivariant representation of features: we start from the requirements of equivariant representation to derive step by step how the structure of CAN and extra speed neurons for realizing translation operators are determined.
>
>  Specifically, the novelty of our study includes:
> - We start from Eq. 5 (Lie group) to derive the Eqs. 21 and 22 (the model of compass circuit) to show the compass circuit is able to achieve equivariant representation.
> - The Eq. 22 is a novel theoretical result that gives a theoretical prediction of implementing path integration. The theoretical result is not presented in earlier work, e.g., Ref. 30-33.
> - The network model in Sec. 4 is directly derived from the Lie group. Although similar model was proposed in Ref. 20 and 28, the earlier work did not study Lie group.
> - We formulate the concrete path integration problem in neuroscience by the concept of commutation in Lie group.
>
> We believe the present work gives us new insight into understanding the well-established CAN dynamics and the Drosophila’s compass circuit, motivates us to apply group theory to interpret more brain functions (see discussions in lines 290-295). We will revise our paper to further clarify and emphasize our contributions relative to past work.
>
> The present work is post-dicting available evidence by using the 1d translation group. In the future we believe that we can use other groups to predict neural circuits (discussed in lines 290-295).
>
> ### Comparison with previous works
>
> Due to the page limit, we did not systematically compare our work with previous works in detail. To our best knowledge, no previous works connected Lie group to a well-established recurrent circuit mode (i.e., CAN), nor a concrete neural circuit in the brain.
>
> Specifically, the network model in Sec. 3 is inspired by Ref. 27, the model in Sec. 4 is similar to Ref. 20 and 28, and the model in Sec. 5 is similar to Ref. 30-33. However, previous works have mainly focused on studying the dynamics of CAN and its roles in brain functions, and have not studied CAN from the point of view of group equivariant representation.
>
> ### Introducing external input to the neural circuit
>
> Our work emphasizes an internal mechanism which actively induces the translation of neural representation (continuous attractors), and hence the external input is not included for the sake of simplicity. We are also interested in introducing external inputs that representing instantaneous stimulus direction and the instantaneous velocity into Eqs. 19a and 19b respectively. Some examples can be found in earlier results (Ref. 30 and 33). Specifically, an external input which has a bump profile located at the instantaneous stimulus direction can be introduced in Eq. 19a. And the external speed input can be introduced to all speed neurons, e.g., Ref. 30 introduced a spatially constant term whose value is dependent on speed to all speed neurons.
>
> From Drosophila’s study, we know that the external input to speed neurons conveys the speed information, which can be computed from sensory inputs or inherited from the efference copy of the motor command. It is also possible that the connections from CAN to speed neurons also help to compute the speed information, but further experimental evidence is needed.

---

> > ### Comment · Reviewer_fomB · 2022-08-07
> > **Response to rebuttal**
> >
> > Thank you for your clarifying the novelty and main contributions of your paper. Reading through the paper again with these clarifications was very helpful and I now believe this is an important contribution. I have raised my score accordingly.

---

> ### Author Response · Authors · 2022-07-31
> **Reply to reviewer's questions (2nd part)**
>
> ### Reply to reviewer’s questions
>
> 1. The missing citation in line 37: Deneve, J. Neurosci., 2007
>
> 2. The derivation of Eq. 7 from Eq. 6 relies on two assumptions as listed in lines 93 – 95. We assume homogeneous neurons in the population (line 93), i.e., every neuron has the same profile of tuning curve, and the preferred stimulus $x_j$ of all neurons are uniformly distributed in the stimulus space. The two assumptions are heavily used in neural coding studies (e.g., Pouget, Neural Computation 1998; Dayan&Abbott, 2001; Wu, Neural Computation 2002).
>
> 3. The $u(x-s)$ is just the change of notation of $\bar{u}(x)$, in order to emphasize the neuronal response depends on the difference between x and s.
>
> 4. The discussion of introducing an external input is presented in the above.
>
> 5. The free parameter s in line 131 means that whatever the value of s is in Eq. 10, $\bar{u}(x-s)$ and $\bar{r}(x-s)$ are the stationary states of Eq. 8a-8b. This indicates that the network holds a continuous manifold of attractors, and the location of each attractor in the attractor manifold is characterized by s. This is a unique property of continuous attractor networks.
>
> 6. Yes,  Eq. 13 is derived from Eq. 5, while exponentiating Eq. 5 gives $\hat{T}(a)=\exp(a\hat{p})$.
>
> 7. The $\tau v \partial_x W_r$ in Eqs. 15 and 16 is replaced by the rightmost term in Eq. 19a. If we set $\tau$ on the leftmost in Eq. 19b as zero, Eq. 19a-b will be mathematically equivalent to Eq. 15 and 16. The introduction of Eq. 19 is to replace the multiplicative modulation on synaptic weight by the multiplicative modulation on firing rate, while the latter, we think, can be more easily implemented by neural circuits and has also direct evidence from the Drosophila experiment.
>
> 8-9. We will fix the typos in the revised manuscript.
>
> 10. The notation $U_j$ in Eq. 6 is not union, but is a scalar representing the peak firing rate (Eq. 92).
>
> 11. We will fix the different definitions in Eqs. 7 and 10. They only differ by a coefficient in the denominator inside the exponential function, and this difference does not change our later results.

---

### Official Review · Reviewer_C6Qo · 2022-07-11

**Rating:** 7
**Confidence:** 4
**Soundness:** 3 good
**Presentation:** 3 good
**Contribution:** 3 good

**Summary:**

Many artificial neural networks should obey symmetry relationships.  This paper derives a translation operator for a ring attractor network inspired by recent findings from the drosophila head direction system.   The resulting system is designed to be equivariant to translation and is at least roughly neurobiologially plausible.


**Questions:**


35: ...explicitly represent...
77-8: ...the above derivation...
92: Not convinced it's strictly necessary that the units have identical tuning curves.  Agree that it's convenient.
115: Divisive normalization...
138: ...disrupted by noise.
171: ...utilize the ...
213: Don't know that much about drosophila specifically, but in general, multiplicative modulation of weights is not challenging in neurobiological systems.  Keywords are ``shunting inhibition'' and ``cortical gain control''.

Would be good to note that there are in fact speed cells in the grid system, e.g., Sargolini et al., (2006).  However their firing rate profiles are relatively complex (see for instance, Dannenberg, et al., 2019, J Neuro; 2020, eLife).

Authors might be interested to compare these results to Tanaka & Nelson (2019, PRE).   In thinking about 3-D transformations, Shepard (1994, Psychonomic Bulletin & Review) may be helpful.



**Limitations:**


The paper is far too sanguine about extending this approach to place cells/grid cells.  This is likely to be much more complicated than one might expect from this paper.  Place cells do not monotonically map the 2-D environment.  Locations nearer to reward locations and boundaries are overrepresented.  In addition, the grid system is much more complicated than a 2-D surface.  2-D position is represented in conjunction with head direction and speed.  Moreover, there appears to be a continuous spectrum of time constants in the speed system (Dannenberg, et al., 2019).


**Strengths And Weaknesses:**

Strengths:
This paper strikes a nice balance between abstractness and specificity.  The math is initiated at a very high level, but is then implemented for a very specific circuit.

The model makes a number of predictions that could be tested with existing technologies.

Weaknesses:
My major concern about this paper is that perhaps something similar was already worked out, perhaps for the grid cell system but I'm unaware of that work.

---

> ### Author Response · Authors · 2022-07-31
> **Reply to reviewer's comments**
>
> Thank you for your insightful comments and valuable suggestions. We will revise our paper accordingly.
>
> ### Comparison with previous works
>
> There were earlier works having similar ideas that a separate population of neurons representing transformations, e.g., grid cells represent the 2d translation of positions (e.g., Stachenfeld, Nat. Neurosci., 2017; Whittington et al., NeurIPS 2018;), but to our best knowledge, none of the previous works used theoretical analysis which directly starting from Lie group to derive a concrete, biologically plausible nonlinear recurrent circuit dynamics.
>
> ### Multiplicative modulation
>
> Thank you for the interesting suggestion that shunting inhibition might be a way to achieve multiplicative modulation. We will consider and discuss this possibility in our revision. Another reason motivating us to consider the multiplicative modulation of speed neurons’ firing rate is from the observation of drosophila. We believe that the multiplicative modulation of firing rate results from inhibitory neurons in the circuit, as a form of gain control widely observed in cortical circuits.
>
> ### Neural variability and heterogeneity
>
> To emphasize the main mechanism and simplify the analysis, we consider homogeneous neurons (lines 93) as well as no internal neural variability in the network. We do acknowledge that real cortical circuits have heterogeneity among neurons and there exists large neuronal response variability. It has been suggested that the CAN is resistant to input noise by removing the noise component that is perpendicular to the attractor manifold (Deneve, Nat. Neurosci., 1999; Wu, Neural Computation, 2002), and then the equivariant representation can be still held. The heterogeneity will break the perfect equivariant representation in recurrent network. One possibility is that the neural heterogeneity and neural noise can help the brain implement sampling-based Bayesian inference (e.g., Orban, Neuron 2016; Echeveste, Nat. Neurosci., 2020), where the instant state of the CAN and the activity of speed neurons represent the samples of head direction and rotation speed, respectively (see more details in our reply to Reviewer 2shN).
>
> ### Generalization to grid cells
>
> Thank you for the useful references, and for pointing out the complexities of place cells and grid cells. We shall add more discussions about place cells and grid cells in the revised manuscript.
>
> A recent study (Gardner, et al., Nature 2021) reported that the population activities of grid cells appear to reside in a 2D torus manifold, which is a finite 2D domain with periodic boundary conditions. CAN based on 2D torus has also been proposed in the literature (e.g., Burak and Fiete, PLoS Comput. Biol. 5, 2009). The 2D torus is the 2D version of the ring (which is 1D torus) that is being studied in present study. The 2D additive group with clock arithmetic (due to periodic boundary conditions) is a Lie group and is still abelian. The grid periodicity may be caused by "rolling" the 2D torus over the 2D spatial domain. Therefore it is possible to generalize our current theoretical framework to the 2D domain to partially explain the properties of grid cells in future work.
>
> Meanwhile, we also notice there is a separate population of grid cells that are more selective to speed (Sun, PNAS 2014). Besides, the continuous spectrum of speed neurons’ time constant might form a basis to explain the temporal process, which may potentially be explained by a temporal scaling group.

---

> > ### Comment · Reviewer_C6Qo · 2022-08-08
> > **Time and scaling**
> >
> > This seems *very* relevant to the discussion of grid cells and should probably be cited in the paper.
> > https://link.springer.com/article/10.1007/s10827-020-00742-9
> >
> > Not necessarily for this paper, but the authors may be interested in work by Lindeberg and colleagues on temporal scaling.  For instance:
> > https://link.springer.com/article/10.1007/s10851-015-0613-9

---

> > > ### Author Response · Authors · 2022-08-08
> > > **Reply**
> > >
> > > Thanks for the reviewer's suggestion. We will definitely comment on the first paper in our revised manuscript.
> > > Also, the temporal scaling is quite relevant to our future work and it is pretty interesting to study the temporal scaling in recurrent dynamics from a scaling group point of view.

---

### Official Review · Reviewer_QibX · 2022-07-12

**Rating:** 8
**Confidence:** 4
**Soundness:** 4 excellent
**Presentation:** 4 excellent
**Contribution:** 4 excellent

**Summary:**

The authors provide an exciting extension to the classic bump circuit model [20]. The work provides a connection between Lie group translations and continuous differential recurrent circuits (or continuous attractor networks, CANs) to demonstrate that CANs can effectively encode a continuous variable shift, such as position. They derive the circuit equations, implement numerical simulations that match the theoretical assumptions, and suggest modifications to more appropriately accommodate known biological constraints.

**Questions:**

1) Implications of the gaussian tuning curve assumption (lines 87-88) – A Gaussian profile is certainly a fair assumption; it is used in a majority of neural sensitivity research. But there’s also good evidence to suggest that at least visually selective neurons (probably others as well) are better fit by leptokurtic tuning curves, which demonstrates an increased selectivity for preferred stimuli [1]. I think this could be covered in a future publication (i.e. is not necessary to implement in the rebuttal phase), but I am curious if you have thoughts on how one might apply the derivation in section 2 of the appendix to a generalized normal distribution? Additionally, does neuron selectivity have an impact when you are considering an infinite set?

2) notation clarifications – I initially ran into some confusion with the notion that I think could have been avoided with a more gentle introduction at the top of section 2.

2.a) There are a lot of conceptually overlapping variables that correspond to the stimulus & its perturbations (s, s’), neuron preferences & their perturbations (x, x’), the addition of an index when referring to individual neurons (x_j), and resultants from perturbing a stimulus (e.g. ‘a’ in T(a)*s = s+a). I understand the need for all of them, but an early exposition on what they are to orient the reader would have helped me.

2.b) I also had confusion coming from switching between a discrete (countable) notation for the neurons and a continuous notation based on the preferred stimulus. As some examples of the former, consider the pictorial description in Fig 2.a; the use of the term “index” in the horizontal axis label in fig 2f; and the use of the subscript j in equation 6. An example of the latter is found in the use of an integral over alternate stimuli in the denominator of equation 8b, and the explanation of an “excitatory neuron preferring stimulus s=x” on lines 112-113. This is explained briefly on lines 121-126, although there you walk back the infinite neuron description and describe it as “a large population of neurons.” I am not suggesting that any of this is wrong, per say, but more recommending that you spend a moment earlier (for example near line 86 or at the top of section 2) to explain what is going on here and when/why you will be flexible with notation.

3) What are the empirical implications of the infinite neuron assumption? I would appreciate a discussion at the end of the paper on what it means to implement such a system with a finite set of neurons. I may have missed this in the paper, but what were the implementation details for the experiments shown in figures 3 and 4? How does the number of neurons (i.e. resolution of the discretization of the stimulus space) influence the analysis?

4) Line 22 “decode the instant orientation” did you mean instantaneous?

5) Typo on line 35 of the supplement – “clear to se”

[1] Ringach, D. L., Shapley, R. M., & Hawken, M. J. (2002). Orientation selectivity in macaque v1: Diversity and laminar dependence. Journal of Neuroscience, 22(13), 5639–5651.


**Limitations:**

The authors note limitations in sections 5 and 6. I note some questions that touch on limitations of the presented work (e.g. the mismatch between theory & experimentation).

**Strengths And Weaknesses:**

**Originality:** Of course the concept of using a recurrent circuit to encode an environment variable, such as position, is not new. Additionally, the application of Lie algebra to achieve equivariance in neural circuits is not new. However, as the authors point out and as far as I know, the connection that they provide between these two concepts is both novel and exciting. I am fairly confident that the additional clarity and rigor that the present study provides over previous work in the space is an original contribution.

**Quality:** I believe the mathematics are technically sound, and the supporting experiments demonstrate the applicability of the theory. I have some minor comments below about including more discussion on weaknesses in applying the theory.

**Clarity:** I include minor comments below to improve clarity, but overall the exposition is clearly communicated.

**Significance:** I believe the submitted work is a significant theoretical contribution. It draws on several ideas from iconic work in neural coding [20], and the application of Lie algebra lends itself to several future extensions.

---

> ### Author Response · Authors · 2022-07-31
> **Reply to reviewer's questions**
>
> We are deeply grateful for your positive review and insightful comments. We shall follow your valuable suggestions to improve the presentation of our paper.
>
> ### 1. About non-Gaussian profile.
>
> This is an interesting point. The profile of tuning curve is determined by the recurrent connection profile (Eq. 9) in the network model. In principle, we can vary the recurrent connection profile, to let the recurrent circuit self-consistently generate different tuning curves.
> In particular, the high kurtosis profile may be considered a scale-mixture of Gaussians, i.e., Gaussians with different scales. And then the recurrent connections could also be set as the mixture of many Gaussian connection profiles to generate the high kurtosis tunings. We shall explore this possibility in future work.
>
> Mathematically, we can use the math trick from the conjugate prior in Bayesian statistics to ensure that the recurrent connections and the firing rates of neurons are in the same family of profiles, e.g., Gaussian or generalized Gaussian.
>
> In the present study, the neuronal selectivity does not matter when there are an infinite number of neurons. But if the tuning curve is monotonic, then the network dynamics will be heavily affected by infinite number of neurons, resulting in that the integral of the monotonic tuning diverges.
>
> ### 2. Improving the notation.
>
> Thank you so much for the detailed suggestions on improving the notation and presentation.  We will follow your suggestions in revising our paper. We will differentiate the notation for discrete and continuous variables clearly in the revised manuscript.
>
> ### 3. About the infinite number of neurons.
>
> There are two main reasons for considering infinite number of neurons in the present study. First, the 1d translation operator is defined to act on a continuous function $\bar{u}(s)$ (Hilbert space, lines 25), which is interpreted as the network response. Second, considering the limit of infinite number of neurons brings a lot of benefit to theoretical analysis, e.g., multiplication of the recurrent connection matrix with neuronal firing rates can be treated as an integration. In the simulation, as long as the number of neurons is not too small, the simulation results can be approximately regarded as achieving equivariant representation. But if the neuron number is too small, the network responses cannot translate along the attractor manifold smoothly and will exhibit the zigzag effect.
>
> The Drosophila’s heading direction circuit contains about 40 E-PG neurons as mentioned by reviewer 2shN. For finite number of neurons, the network model can still have equivariant representation, except that the profile and the convolution kernel may slightly deviate from Gaussian. And the profile and kernel may be obtained by numerical optimization, as suggested by recent work (Norman et al., bioRxiv 2022). Note that Norman’s study did not formulate from group equivariant representation point of view.
>
> We shall discuss this point in the revised manuscript.
>
> ### 4-5 typos
> In the line 22, it should be the “instantaneous orientation”. We will correct the typos in the revised manuscript. Thank you!

---

> > ### Comment · Reviewer_QibX · 2022-08-10
> > **Response to rebuttal**
> >
> > Thank you for your responses. I appreciate the additional discussions that the authors have promised. It is clear to me that this submission is an important contribution.

---

### Official Review · Reviewer_2shN · 2022-07-15

**Rating:** 6
**Confidence:** 4
**Soundness:** 4 excellent
**Presentation:** 4 excellent
**Contribution:** 3 good

**Summary:**

The paper introduces a translation-equivariant neural model where the neural response to a stationary stimulus is equivalent to convolving with a Gaussian; however, through recurrence, the response is maintained in the absence of a stimulus.  Introduces the use of Lie algebra for deriving the translation generator.




**Questions:**

I am curious about challenges the authors faced - i.e., what did not work?


**Limitations:**

general limitations, see above.
societal impact: n/a


**Strengths And Weaknesses:**

Strengths:
- the framing in terms of group theory, equivariant representations, and the exposition is fairly clear.

Weaknesses:
- it is unclear to me what parts of the paper are mostly for exposition (drawing on previous work of Kechen Zhang), and what parts are novel contributions. Most of the paper consists of theoretical derivations, and it’s often not made clear where results were proved in other papers.
- given the focus on the 1D case I think the title is a bit too general.

For example:
- Section 3.2 shows that the fixed points of continuous attractors are equivariant representations, but I’d be surprised if this result was not already known or demonstrated
- Section 4.1 uses the derivative as the translation operator, as does Kechen Zhang’s (1996) paper which is cited here but not really discussed. It seems that the real contribution is the connection between this derivative to group actions, but I wish this were more clearly laid out.

And some thoughts on the circuit model:
- The discussion of biological (im)plausibility seems weak. There’s not really a problem of multiplicative modulation — synaptic weight multiplied by variable input would do fine (as the authors do in eq 19b). And the other problem they mention — Dale’s law — is easily circumvented by having excitatory and inhibitory neurons, again as the authors do.
- The more critical biological concerns, I think, are (1) biological noise (spike timing, synaptic noise, etc.) and (2) that the Drosophila E-PG cells number about 40, whereas the attractors considered are in the infinite neuron limit. I think some acknowledgement of these points/experiments proving the analysis holds up under these constraints would strengthen the paper.
- in particular there’s a fair amount of literature with CANs and how they deal with noise, drift, limited precision (e.g. Burak & Fiete, 2009) — and it’s surprising to see very little of this discussed here.

Given that velocity-controlled inputs are already the inputs to many CANs for path integration, and examples beyond integrators aren’t discussed here, it seems the main contribution is an interpretation of/twist on existing models, rather than a new model per se. For this reason, I would have appreciated more suggestions on what to do with this Lie group understanding: whether on novel neural circuits, new neural mechanisms that implement the transport operator, or even learning of group of structure.

---

> ### Author Response · Authors · 2022-07-31
> **Reply to reviewer (1st part)**
>
> Thank you for your insightful comments and helpful suggestions on improving the presentation of the paper. We shall revise our paper by following your suggestions.
>
> ### On the novelty and main contributions of the paper
>
> The novelty and main contributions of this paper are that it provides overarching connections between Lie group, continuous attractor network (CAN), and the drosophila’s neural circuit, and demonstrates for the first time how recurrent neural circuitry in the brain implements equivariant stimulus presentation. Here, we present a new perspective from the Lie group theory to understand CAN as a way for the brain to implement equivariant representation of features: we start from the requirements of equivariant representation to derive step by step how the structure of CAN and extra speed neurons for realizing translation operators are determined.
>
>  Specifically, the novelty of our study includes:
> - We start from Eq. 5 (Lie group) to derive the Eqs. 21 and 22 (the model of compass circuit) to show the compass circuit is able to achieve equivariant representation.
> - The Eq. 22 is a novel theoretical result that gives a theoretical prediction of implementing path integration. The theoretical result is not presented in earlier work, e.g., Ref. 30-33.
> - The network model in Sec. 4 is directly derived from the Lie group. Although similar model was proposed in Ref. 20 and 28, the earlier work didn’t study Lie group.
> - We formulate the concrete path integration problem in neuroscience by the concept of commutation in Lie group.
>
> We believe the present work gives new insight into understanding the well-established CAN dynamics and the Drosophila’s compass circuit, motivates us to apply group theory to interpret more brain functions (see discussions in lines 290-295). We will revise our paper to further clarify and emphasize our contributions relative to past work.
>
> ### Comparison to previous works
>
> Due to the page limit, we did not systematically compare our work with previous works in detail. To our best knowledge, no previous works connected Lie group to a well-established recurrent circuit mode (i.e., CAN), nor a concrete neural circuit in the brain.
>
> Specifically, the network model in Sec. 3 is inspired by Ref. 27, the model in Sec. 4 is similar to Ref. 20 and 28, and the model in Sec. 5 is similar to Ref. 30-33. However, previous works have mainly focused on studying the dynamics of CAN and its roles in brain functions, and have not interpreted CAN from the point of view of group equivariant representation.
>
> ### On finite number of neurons
>
> Thank you for pointing out this important issue. Indeed, there are only 40 E-PG cells in the Drosophila head-direction circuit.  For the convenience of theoretical analysis, we have considered an idealized CAN with infinite number of neurons,  resulting in that the translation operator (Eq. 1) acts on a continuous function (Hilbert space, lines 25).
>
> For finite number of neurons, the network model can still have equivariant representation, except that the profile and the convolution kernel may slightly deviate from Gaussian. And the profile and kernel may be obtained by numerical optimization, as suggested by a recent work (Norman et al., bioRxiv 2022). Note that Norman’s study didn’t formulate from group equivariant representation point of view. We will discuss this point in revision.
>
> ### On the neuronal response variability
>
> To highlight the basic mechanism of implementing equivariant representation, the current study has not included neural noise. It has been suggested that the CAN is resistant to input noise by removing the noise component that is perpendicular to the attractor manifold (Deneve, Nat. Neurosci., 1999; Wu, Neural Computation, 2002), and then the equivariant representation can be still held.
>
> Apparently, the noise will break the perfect structure of equivariant representation in a CAN. It has been suggested that the internal noise is essential for the brain implementing sampling-based Bayesian inference (e.g., Orban, Neuron 2016; Echeveste, Nat. Neurosci., 2020), where the instant state of the CAN and the activity of speed neurons represent the samples of head direction and rotation speed, respectively. There were recent studies suggesting that a CAN with internal Poisson-like variability performs sampling-based inference on the attractor manifold (Zhang, et al., bioRxiv 2020).

---

> ### Author Response · Authors · 2022-07-31
> **Reply to reviewer (2nd part)**
>
>
> ### On the multiplicative modulation
>
> We are concerned that whether multiplicative modulation directly on synaptic weights is biologically plausible, and whether this modulation is fast enough to match the neuronal dynamics. Therefore, we consider multiplicative modulation on the firing rate of speed neurons, which is supported by the Drosophila’s experiment. In reality, multiplicative modulation of neuronal firing rate can be done by inhibitory neurons in the circuit, in the form of gain control as widely observed in experiments.
>
> ### On the title of the paper
>
> Based on your comment, we would like to narrow down the title of the paper to be “Translation equivariant representation in recurrent networks with a continuous manifold of attractors”. In the brain, there exist 1D CAN (orientation, head direction, etc.) and 2D CAN (spatial location), and rarely 3D CAN. Our results can be extended to 2D CAN.
>
> ### Reply to the reviewer's question about the challenges the authors faced.
> There are several challenges we are facing to extend our current work. For example, our work assumes the infinite number of neurons to simplify the theoretical analysis, and it is difficult to obtain closed-form solutions for the finite number of neurons, although numerical approaches work (see our detailed reply above). Moreover, our work does not consider noise and neural heterogeneity and it is a fundamental question to study how the equivariant representation is compatible with those features (see our detailed reply above). Besides, we are still working on extending our method to explain grid cells and place cells, and/or develop a biologically plausible recurrent circuit model to represent a *non-commutative* group structure.

---

### Meta-Review · Area_Chair_Yv4M · 2022-08-26

**Recommendation:** Accept
**Confidence:** Certain

**Metareview:**

The paper constructs recurrent neural circuits that represent stimuli equivariantly with respect to a given symmetry, taking the example of the 1D translation group. Most Reviewers were positively impressed by the general framing of the problem in terms of group theory and the elucidation of a connection between Continuous Attractor Networks and Lie groups.
The main weakness pointed out by the Reviewers was that the high-level exposition of the paper tended to conceal the distinction between novel original contributions and the previous literature. This concern was however resolved in the rebuttals in a way that seems to satisfy all Reviewers.
Further concerns about the biological plausibility of the proposed general construction and the potential difficulty of extending it beyond the restricted 1D case analyzed in the paper were also raised by several comments in the reviews, but also in this case the clarifications in the rebuttals seemed to convince Reviewers, who by and large expressed optimistic views about the significance of the work for its potential for future developments and applicability for modeling other neural systems.

**Award:**

No

---

### Decision · Program_Chairs · 2022-09-14

Accept